# Sonorous Touches: Listening to Jean-Luc Nancy's Transimmanent Rhythms

Adi Louria Hayon 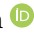

Art History Department, The Yolanda and David Katz Faculty of Art, Tel Aviv University, Tel Aviv 6997801, Israel; adilouria@tauex.tau.ac.il

**Abstract:** Luigi Russolo's *Intonarumori* together with his manifesto *L'arte dei rumori* (1913) marked a break with the art of clear signification. From here on, noise and dispersed sounds replaced the concept of music reverberating the harmony of the spheres by propelling the quandaries of immanence contingent on palpable resonance performing the differential relational manner of heterogeneous existence. This somatic turn is central to Jean-Luc Nancy's *Listening*, where he proposes listening as a tangible fundamental resonance rumbling the *corpse sonore*. This paper elaborates on the move from the art of music to the plurality of rhythmic worlds. Nancy's proposition of sonorous existence demonstrates two movements, one that retreats from hearing the Pythagorean musical-arithmetical cosmos exhibited in Robert Fludd's *Monochord*, the other plays the singular plural pulsations of dispersed creation performed by Michael Snow's *Tap*.

**Keywords:** Jean-Luc Nancy; Luigi Russolo; Robert Fludd; Michael Snow; aesthetics; speculative music; sound art; hapticity; continental philosophy; sonic materialism; immanence; heterogeneity; empiricism

> *"What's this* this, *who is the body? This, the one I show you, but* every *"this"? All the uncertainty of a "this", of "thesis"? All That? Sensory certitude, as soon as it is touched, turns into chaos, a storm where all the senses run wild."*
>
> **Nancy,** *Corpus*

Luigi Russolo's *Intonarumori* (noise-tuners) together with his manifesto *L'arte dei rumori* (The Art of Noise, 1913) marked a break with the sounding art of clear signification. "We must", he wrote in his manifesto, "break at all costs from this restrictive circle of pure sounds and conquer the infinite variety of noise-sounds... Although the characteristics of noise is to brutally bring us back to life, the art of noises must not be limited to a mere imitative reproduction..."(Russolo 1986, p. 25).[1] From here on, noise and dispersed sounds replaced the concept of music reverberating the harmony of the spheres to propel the quandaries of immanence in construing an ungrounded manner of heterogeneous existence. Russolo's immediate call to lend an ear to noise is a double brutality, the first looks out to cut the chord of celestial harmonies; the second, brings us back to life. It is a call for immanent art, which stemmed from the Futurists fierce and violent political ideology. However, Russolo's brutality did not merely operate in the politics of the pre-WWI days, it was a revolutionary manner seeking to do away with a particular kind of *mimesis*, binding the nature of music and philosophy, that is, the metaphysics of *Harmonices Mundi* from which infers the diagrammatic world demonstrated in an arithmetical tonal system, systematic notations, and musical instruments. In its place, the Futurists returned to life. They proposed avant-garde art practiced within a somatic belief that existence is an eternal becoming in flux contingent on the sensations of proximate bodies.[2] Consolidating these new ideas in the art of noise, Russolo together with Ugo Piatti, built new sound machines (noise-tuners, *Intonarumori*), that echoed an attack on the tuned audibility of the

concert hall ([Brown 1981–1982](#), pp. 31–48). On 1 March 1914, he published in the Italian magazine *Lacebra* his first graphic representation of "dynamic continuity" in the musical enharmonic notation "Ii risveglio di una città" (The Awakening of the City). The notation exhibiting Russolo's thinking summons visual representation into a new graphical scheme hung mid-air between a conceptual proposition and a musical score guiding performance ([Vattano 2021](#), pp. 55–56).

In 2002, almost a century after Russolo's sonorous revolution, Jean-Luc Nancy attempted to elucidate the difference between hearing and listening, that is, between posing, imposing, projecting, and morphing pregiven forms, and sonorously sensing that which outweighs form, enlarges it, penetrates it, agitates its tuning cords by "*touching*, putt[ing] into play the whole system of the senses" ([Nancy 2007a](#), p. 3). Hearing for Nancy, adheres to philosophizing logically; listening is a proposition for considering "resonance as a foundation, as a first and last profundity of 'sense' itself (or of truth)" ([Nancy 2007a](#), p. 6). And it is my perceptual experience, my senses, the stretch of my ear, that makes sense. Each time singular, eventful, relational. In this sense (what is sense? We will return to this), Nancy takes on the challenges of empiricism by which experience, my singular experience, becomes the a priori condition for the possibility of experience. "But the challenge in a study of the senses and perceptible qualities", he writes,

> "is necessarily the challenge of an empiricism by which one attempts a conversion of experience into an a priori condition of possibility. . . of the experience itself, while still running the risk of a cultural and individual relativism, if all the "senses" and all the "arts" do not always have the same distributions everywhere or the same qualities". ([Nancy 2007a](#), pp. 10–11)

When Nancy writes of converting experience into an a priori condition of possibility, he weakens the superiority of reason and rejects metaphysical causality. Therefore, the possibility of making sense does not consider the perceptible qualities of the sensual senses secondary to cognizant world-forming, but are effectively and performatively relational to thought. Thus staged, sense is a composite term for Nancy. Sense may denote meaning, the sensual senses, or a manner consolidating a *telos* toward signification, that is, a manner toward an image to come. In *The Creation of the World or Globalization*, he ties the activity of making sense to world-forming (*mondialisation*), which is the creation of a world ([Nancy 2007b](#), p. 52). Hence, when probing a specific sense, the audible, the challenge of empiricism to convert experience into an a priori condition always steps out of sensual sense toward making legible sense which exposes the condition of the possibility of meaning, endeavoring to articulate an event. The relation between these senses is relative for Nancy since it is a manner of composing an image confined to the conditions giving them rise. If something is relatively so, then there is no one other a priori autonomous set founding the thing. This kind of relativism has no ready consensus on any one definition. A specific eventful experience is contingent on and differs from an underlying variable. We may therefore infer that the a priori conditions being exposed are a kind of contingency on the variable differences of the senses and the arts devoid of pregiven communalities posed beneath or above. Relative contingencies bring forth a dynamic sense exclusive of absoluteness. In straining toward possible sense contingent on the mutual intricacy of the difference between the senses and the arts, Nancy opens the patent distribution of probable sense. In musical terms it is a heterogeneous discordant enharmony that comes to bare the principle of the singular-plural which is "*the difference in sense* (in the 'perceived' [*sensé*] sense of the word) [that] *is its condition, that is, the condition of its resonance*" ([Nancy 2007a](#), p. 11). This paper addresses the heterogeneous nature of Nancy's sonorous proposition contingent on the haptic syncopated difference between listening and hearing, resounding bodies, and intelligible morphing. It is an artistic necessity for making sense as a manner directed toward an image to come and at the same time a mode of existence that lends to the differential dispersed properties of an acoustic expanse. In *Listening*, this proposition accelerates speculative philosophy in probing the tangible relations between the sonorous and its musicality. The musicality of the sonorous, fleeting as it might be, drives a return

that performs the difference which at the same time articulates in auto-reflection while each time resonates differently. Nancy's text is dedicated to this return, and yet, while the philosophical arguments reside in constructing an ontological tonality of differing resonances, the text is perplexingly imbued with musical references from Stravinsky through Wagner, Jazz, Rock, Classical rendering, electronic, computer music, popular, religious, and music of all continents.[3] I consider these references perplexing since Nancy's selections are of the highly articulated, even canonized, musical genres. They pose a distinction between the aesthetic experience of the senses and that which pleases the tuned, even reasonable, ear, only to retreat their relational creative potential. They also pose a quandary about Nancy's style of writing, his *Darstellung.* The text introduces musical references that stand against the dispersed sounds of the *corpse sonore*. In other words, the sonorousness of the text speaks two tongues: it reflects given musical forms while at the same time endeavors to conceptualize the *corpse sonore* through the properties of unorganized sound. It is by this difference that Nancy offers to maintain the prolific relation between the two manners, that of seeking to attain knowledge and that of creating a world and a self. Both are contingent on the production of rhythms and their audible, haptic, and visual experience.

In what follows, we tune to the performance and concept of sonorous-haptic dispersion as it adheres to speculative music and the empirical condition of art and philosophy. While *Listening* is a late text in Nancy's oeuvre, the sonorous ontology tying tangibility with heterogeneity resonates with his early writing where the philosopher touched on the reverbs of delimitation in Kant's faculties.[4] Nonetheless, it is his later pulsations conditioning the originality of the arts and their unbounded senses that bring forth haptic-soundings, sonorous touches. In his philosophical writings Corpus (1992), the opening essay of *The Muses* "Why Are There Several Arts Not Just One?" (1994), and Listening (2002) demonstrate a speculative materiality which touches the relational discords resounding bodies and thoughts. Nancy's mastery of writing is risky. His use of sonorous terminology rumbling the *corpse sonore* together with his rejection of the violence of vision and the truth of *alētheia* may allude to forgetting the schemes of legible cadence while proposing some self-sufficient materiality, and yet, he does not discard signification as fleeting as it might be (Nancy 2007a, p. 3).[5] Therefore, the sonorous body is inherently tied to music as much as to tangible resonance. If *Listening* resonates on, in, and through the differential verges composing the sonorous body mosaic, they penetrate and dissolve articulated cadences, touching internal and external distances that knot a necessary relation between the audible with the haptic. Hence, we may ask whether these two senses carry the same weight for Nancy, or perhaps, the sonorous musical body grounds a metaphor that belongs to Nancy's philosophy of touching in a distance. Disclosing these vibrating rhythms posits Nancy as the philosopher of the origin of heterogeneity and the exposer of the heterogeneous origin of sense, of the arts, and of creation.

Throughout the text, *Listening* references music. In order to understand why and how Nancy repeatedly calls attention to music, we first tune to the difference between hearing and listening as it probes the shift from musical harmony of coherent measure to dispersed experience. Second, we see how making sound with the body propels sonic environments contingent on syncopated relationality. Third, we see why the sonic relations between bodies are necessarily tangible. In what comes ahead I first tune to Nancy's sonorous body in light of the art of speculative music, and then move to his haptic proposition alongside artworks that make sonorous topologies by performing somatic projections. To practice this proposition in art, I first look at Robert Fludd's 17th century illustrations belonging to the representation of speculative music and the human sensing mechanism. These demonstrate the logical sense of music figuring the origin of the world in symbolic diagrams. I then tune to *Tap* by Michael Snow dated to 1969 and exhibited in 1970 at the 25th Venice Biennale. *Tap* performs multiple cadences contingent on staging multi-media and multi-sensorial activation that propel perceptual and somatic effects, destabilizing while at the same time creating new relations between the self and the world. The piece does not map a univocal or rhetorical representation, but produces new effective models for world-forming.

Set between an installation and a performance, *Tap* exposes its technological extensions composed of a black and while photograph, a framed typewritten text, a tape player, a sound tape, a speaker, and a wire. These material properties assume an eventfulness not merely designed to show its own working, but substitutes the ideational regime of representation for instable sonic topologies contingent on material relationality. By producing sounds with the body and staging the technological instruments of such a production, Snow stages the possibility of dispersed multiple rhythms as a proposition for worldly somatic and virtual-cognitive world-forming processes. These physical soundings belong to artistic practices of the 1960s, a time of experimental art that performs internal and external dispersed relations by tapping, pounding and resounding immanently.[6] Snow, and later Nancy, pose the experience of the perceptible qualities as the only manner for interrupting the phenomenological tradition of haptocentrism by dispersing the possibility of legible assimilation while performing the sonorous movements shifting the grounds of sense.

**Hearing/Listening.** In *Listening*, Nancy calls out the limits of philosophy posed by a debilitation in philosophizing contingent on hearing. "Assuming", he writes in the opening paragraph,

> "that there is still sense in asking questions about the limits, or about some limits, of philosophy (assuming, then, that a fundamental rhythm of illimitation and limitation does not comprise the permanent pace of philosophy itself, with a variable cadence, which might today be accelerated), we will ponder this: Is listening something of which philosophy is capable? Or—we'll insist a little, despite of everything, at the risk of exaggerating the point—hasn't philosophy superimposed upon listening, beforehand and of necessity, or else substituted for listening, something else that might be more on the order of *understanding*?" (Nancy 2007a, p. 1)

Nancy's unease with imposing legible hearing upon listening belongs, however critically, to the long history probing a metaphysical causality in speculative music which is unwanted in avant-garde and experimental appearances of speculative materialism attempting to practice a multiplicity of co-existence and possible worlds. Nancy's cautious call to overturn the primacy of a coherent mind sketching out truth in diagrams in favor of giving ourselves to experience as the only mode for exposing being as such, dwells in dissolving the fixed relation of speculative music to truth. He does not merely want to dissolve such relations, but employs the elements of its residue to offer a quasi-transcendental proposition of being-in-the-world contingent on the vibrations of listening without a priori belief, prejudice or judgment. "Isn't the philosopher", writes Nancy,

> "someone who always hears... but who cannot listen, or who, more precisely, neutralizes listening within himself, so that he cannot philosophize?

> Not, however, without finding himself immediately given over to the slight, keen indecision that grates, rings out, or shouts between "listening" and "understanding": between two kinds of hearing, between two paces [*allures*] of the same (the same sense, but what sense precisely? that's another question), between a tension and a balance, or else, if you prefer, between a sense (that one listens to) and a truth (that one understands), although the one cannot, on the long run, do without the other?" (Nancy 2007a, pp. 1–2)

For this reason, *Listening* touches upon the esoteric philosophy of speculative music, which provides Nancy with the possibility of critiquing a peripheral discourse right at the heart of the empirical challenge. By playing with the dual hearing/listening Nancy does not discard the first, but proposes a particular fold of which one leg is grounded in the sensual sense of listening and the other in the fabulous metaphysics of hearing. Nancy dwells in between the two, at the vibrating limits of speculative music resounding self and world. It is with such limits that he opens the text in order to position his philosophizing activity as an investigation that resides in a particular historical current, while at the same

time a relational contour, a delineation of limits, that is a manner or mode of creative exposition.

Limits for Nancy are relational-differentials, at once separating, syncopating, and connecting. They may mark projected isomorphism or they may become vital resonating reverberations. Limits separate and connect hearing and listening as if posing once again the *meta*physical problem in a co-existential analytic of being-with. Hearing is conceptualized as the power of understanding which neutralizes listening in order to contemplate ideas, to theorize intelligibly, to legibly morph models and meanings. Listening on the other hand, does not make evident the truth of a phenomena, but is a transitive and continuous alteration of referrals, at once spreading out in proximity and penetration of external limits while resounding internally. According to Nancy, if we were able take the risk of exposure by withdrawing from our habitual tendency to make sense logically or to produce coherent meaning, we would find that at the bottom of hearing is listening. It is a fundamental resonance and a resonance as a foundation, that accepts Pythagoras sonic foundation without its univocal origin or the thoughtful *Quadrivium* manifesting the arithmetic classification forming the world and its movement, for it belongs to the ontology of being-with contingent on singularity defined by alterity and the register of feeling (not thinking).[7] By concentrating on listening as feeling, Nancy overturns the dominant superiority of thinking to offer a manner of philosophizing contingent on somatic performance such that its main manners of being in the world are contingent of the singular differential relations of bodies echoing, resonating, and reverberating. This sonic-somatic proposition exchanges the means of the philosopher and the artist in a manner that "form, idea, painting, representation, aspect, phenomenon, composition", are replaced by "accent, tone, timbre, resonance, and sound" (Nancy 2007a, p. 3).[8]

**Measured Hearing and Speculative Music.** Nancy's imperative ocular/audial division continuously stages spatio-visual formations that do not forget the long history of speculative music. Roger Grant positions *Listening* as a musicological counterpart of Nancy's writings on the visual and written arts (Grant 2009). In "Why Are There Not Several Arts", Nancy separated the arts from one origin. Almost a decade later, *Listening* is an enquiry that should be positioned at the proximity of the plurality of arts which should not be deduced from Nancy's thoughts about the plurality of senses. The text's critique of hearing does not involve an investigation of sound but speaks about music.[9] Sarah Hickmott finds a difficulty in Nancy's musical language which may infer a confinement to a discipline that imposes and clogs his call for an open somatic organism. She illuminates how the *corpse sonore* draws from the theoretical treatise of composer and theorist Jean Philippe-Rameau *Nouveau système de musique théorique* (1726) (Hickmott 2015, pp. 484–85). Hickmott's decisive critique vocalizes Adrienne Janus's bewilderment as to the disjoint nature running between Nancy's sonorous turn to sounding the flux of multiple bodies while keeping a style of writing and lingo imbued with musical references.[10] While the philosopher writing at the cusp of a new millennia could easily reference experimental music or organized sound, he continuously returns to music. *Listening* does indeed speak of music, musical history, and musicology, but since we are in the register of empirical relativism, Nancy detaches the tyranny of universal music by considering musical manifestations are each time a singular manner of making sense. If we look at the long history of speculative music, we find articulated significations hung mid-air between metaphysical propositions and musical worlds. Since Nancy's sonorous being is a quasi-transcendental proposition that cannot forget language or representation, he may be critical of their superimposing power, but he does not reject them as propositions since they expose a manner of a specific experience and a singular creation of a world. Music is a language, and as Mirt Komel suggested, Nancy is a philosopher of language, a sculptor of metaphors, and yet his are *transimmanent* formations that perform the limits of bodies without end (Komel 2016, pp. 119–20, 124). "A 'transimmanence'", Nancy writes in "Why Are These Several Arts?", "Art exposes this. Once again, it does not 'represent' this. Art is its ex-

position. The transimmanence, or patency, of the world takes place as art, as works of art" (Nancy 1996, pp. 34–35).

Nancy's critique of hearing belongs to his critique of vision maintaining a rejection of an a priori organized perspective carving his way to the heterogeneity of the *corpse sonore* contingent on transimmanence. Posing Nancy's sonorous body in light of the art of speculative music reveals the difference between sense, the art of music, and technicity. The strongest confinement to listening belongs to musical delineation imposing a visual construe contingent on metaphysical causation. Hence, in order to understand what he rejects, we need to tune an ear and focus our eyes to the history of speculative music. Speculative music is a classical approach to understanding the cosmos in musical terms as they are reflected in our minds while harmoniously dwelling in the world's song. Its schematic logic derives from the impossibility to attain unmediated knowledge; therefore, the musical world becomes legible most dominantly by the symbolic representation of arithmetic.[11] Hence, speculative music speculates, that is, it mirrors the natural laws of the cosmos as if it were an arithmetical machine hung in mid-air.[12] On the one hand, it looks out toward the metaphysical realm of ideas in order to gain its forms while emancipated from matter; on the other, it effects the empirical formations of the physical world. For Boethius, as Joscelyn Godwin reminds us, speculative theorists are considered first among musicians since they alone sought to understand the laws and nature of the cosmos *sub specie musicae*, that is, through the categories of music (Godwin 1982, p. 374). They are also first practitioners as they are able to arouse feelings by producing musical instruments and making music that completely adheres to their science. Hence, speculative music links aesthetics to physics, establishes universal and relative validity, and offers both an understanding of the cosmos while arousing feelings of belonging to the world. Throughout the 20th century, musical theorists, historians, and ethnographers from Albert von Thimus to Hans Kayser have maintained the founding schemes of Pythagorean harmonics theory of proportions based on numbers to show their reflections in what they considered as an empirical understanding of the self, the world, and art (Godwin 1982, pp. 373–89).

As noted above, Hickmott's commentary on the *Listening*'s sonotropism rests on the idea that since Plato, music has held the privilege of being a non-mimetic art that embodies a metaphysical power, which as Martin Scherzinger asserts, is not subsumed under the mediator of representation (Hickmott 2015, p. 481; Scherzinger 2012, p. 345). Nonetheless, musicalized imaginary belongs to a philosophical thinking, and I would add artistic practice, of pattern formations. The internal conflict of theorizing excessive stages is a constant tension in any representational proposition, including that of Nancy's writing. Hickmott then positions the *corpse sonore*, drawing both the term and the fundamental bass resounding its particular arithmetic portrait from the theoretical treatise of the composer and theorist Jean Philippe-Rameau *Nouveau système de musique théorique* (1726) (Hickmott 2015, p. 484).[13] Philippe-Rameau's fundamental resounds the metaphysics of reason under the influence of Descartes. If we return to the intention to convey the world's harmony through the science of arithmetical principles, the age of reason reserved much of the structures in debt to previous Western musical-philosophical currents. A case in point is Robert Fludd's illustration of the world-harmony. In 1617 the English Paracelsian physician illustrated the origin of the world and the raison d'être of the hearing world. In *Monochordum Mundi*, Fludd illustrated the cosmic world as a divine chordophone whose spherical body is disclosed by a geometrical diagram determining the intervals of the planet (Figure 1) (Ammann 1967, pp. 198–99). The illustration shows a divine upper hand tuning the world's fundamental string from which musical chords are delineated and played. Following the Pythagorean *Music of Spheres*, the world's body is determined according to the close association of music, mathematics, and geometry (Heller-Roazen 2011, pp. 14–16). The intervals between the planets' orbits correspond to the distance between pitches on one string. Pythagoras' harmonious cosmos recurs in the dialogue of the *Timaeus*, in which Plato describes the creation of the world in mathematical technique. The world is created by the Demiurge, who sculpts the universe into a harmonious structure consisting of seven

scales made up of intervals. By deduction, the complete structure of the physical world is derived from perfect forms, that is, ideas. As the world is based on whole and perfect ideas, it references the intervals of musicosmology that measure space and the movements of bodies.[14] Pythagoras and his followers noted that the relation between musical pitches may be expressed in terms of numerical ratios; the octave as 2:1, the fifth 3:2, the fourth 4:3, and the whole tone 9:8. In accordance to these ratios, we find that similar proportional relations govern the structure of the created world. From this proportioned scale, chronological time and geometrical space are generated as echoes of an a priori diatonic scale, a sonic fundamental. The proportioned parts of the cosmos are rendered harmonious through their participation in a delimited macroscopic whole. Whatever derives from this whole is subjected to the repetition of the same.

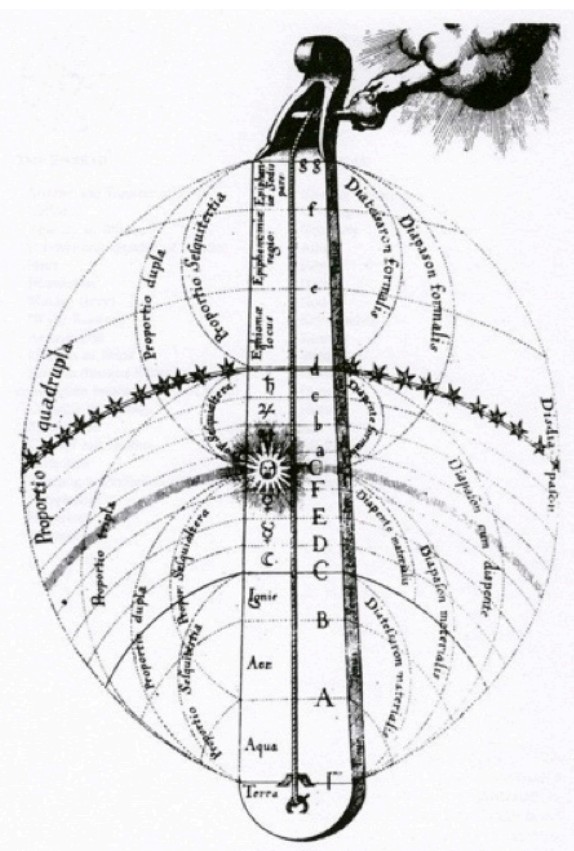

**Figure 1.** Robert Fludd, "Monochordum Mundi" (sonification of the world order with the monochord), *Utriusque mundi. . . historia*, i, Tract I, 90 (pp. 200ff., 215), 1617, paper engraving, 26.3 × 17 cm. Wellcome Collection. Public Domain.

Pythagoras' musicosmology is also shown in Fludd's *De integra microcosmi harmonia* (Figure 2, 1619) from which we learn that the substratum of the cosmos is a musical scale composed of the world's mechanism and instrumental bodies. Such bodies adhere to the very ground that generates their idiosyncratic differentiation. In symphonic composition, the human body and its perceiving sensual capacities are subjugated to the concatenation of creation from *Mundus Intellectualis* (God), to *Mens Intellectus* (ratio), which then branch out to incorporate *Mundus Imaginabilis* (the world of imagination) and *Mundus Sensiblis* (the world of the senses). Defeated by the intellect, the five senses attempt to sense the four substances of the world. Earth is associated with tactility, taste and smell with water, light with sight, and audition with thin air. The spacing at the proximity of *Auditus* and *Visus* marks the liminal locus separating the two senses. This liminal differential is ordinarily thought of in passivity subsumed under the categories of the senses. It is an interval

residing between the senses that marks an apodictic and functional set limit. Critiquing the causality of this legible model, Nancy opens several intervals that are now considered as active spacings of creation: these are the gaps which reside between the singular sensual senses and the opening between the sensual senses and the regulating mind. Characteristic of Nancy, who we may think of as the philosopher of the syncope, such syncopations are necessary for installing the differences and the relations between the senses and the arts as the premises of heterogeneity and creation. In order to weaken and flex the set intervals rationalizing the music of the spheres, Nancy first rejects their categorial imposition and the truth of their causation. He seeks to release both in order to argue that the difference between the senses is not a fixed locale but the creative spacing of their sensual relations from and to self:

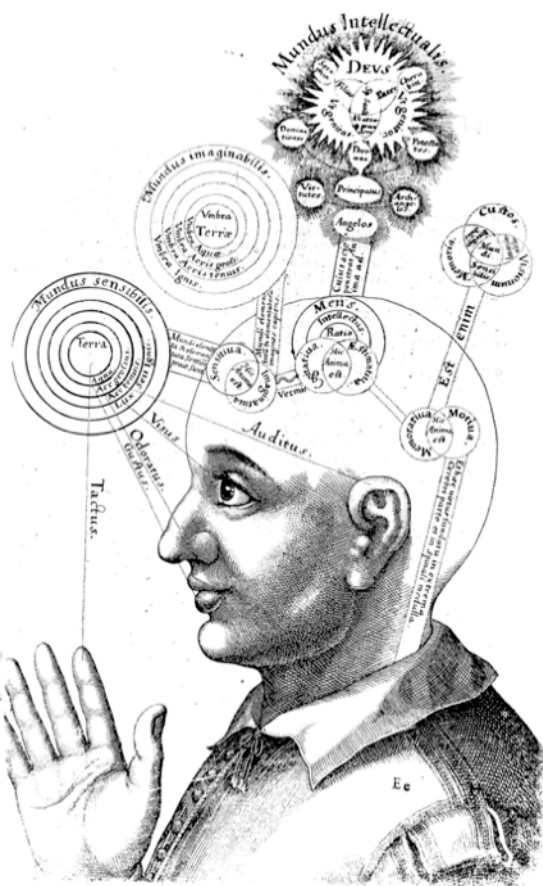

**Figure 2.** Robert Fludd, "Human Mind and Consciousness", *De integra microcosmi harmonia*, liber X (p. 217), 1619, paper engraving, 27.94 × 17 cm, Wellcome Collection. Public Domain.

> "To be listening is thus to enter into tension and to be on the lookout for a relation to self: *not*, it should be emphasized, a relationship to 'me' (the supposedly given subject), or to the 'self' of the other (the speaker, the musician, also supposedly given, with his subjectivity), but to the *relationship in self*, so to speak, as it forms a 'self' or a 'to itself' in general, and is something like that ever does reach the end of its formation. Consequently, listening is passing over to the register of presence to self, it being understood the 'self' is precisely nothing available (substantial or subsistent) to which one can be 'present,' but precisely the resonance of a return [renvoi]. For this reason, listening—the opening stretched toward the register of the sonorous, then to its musical amplification and composition—can and must appear to us not as a metaphor for access to self, but as the reality of this access, a reality consequently indissociably 'mine' and 'other,' 'singular' and 'plural,' and



much as it is 'material' and 'spiritual' and 'signifying' and 'a-signifying'". ([Nancy 2007a](), p. 12)

Nancy overturns the Platonic origin of the world for several reasons: The first is to do away with causation, that is with the source of perfect forms. "That 'the body'", he writes in *Corpus*, "might serve as the name for the Stranger, absolutely, is an idea we have pursued to its successful conclusion... Above all, let's not act as if this thought hadn't been projected across the picture for a long time... (Maybe—maybe this word could be saved by beautiful geometrical designs in three or *n* dimensions, with elegant axonometries: but then everything would have to float, hanging in mid-air, and bodies *must* touch the ground" ([Nancy 2008a](), p. 9). The second, is to release speculative diagrams from deductive methods and elemental taxonomies. Speculative diagrams are released to be hung mid-air. The multiple arts, and the art of music, produce each time a technique. It is an artistic decision that is at once a mediation and syncopated externality. "Technique means knowing how to go about producing what does not produce itself by itself. Technique is a—perhaps infinite—space and delay between the producer and the produced, and thus between the producer and him- or herself. It is production in an exteriority to self and in the discreteness of its operations and its object. In this regard, the singular plural of art extends to the endless multiplying of the artist's technical decisions" ([Nancy 1996](), p. 25). Third, he releases the senses from their inferior compartmentalization and their monomorphic purposiveness by importing Freud's energetic model of erogenous zones. In "Why Are There Several Arts?" Nancy uses these sensual zones for their discrete diversity, for their mobility, and for their potential to propose sensual excess that interrupts the stasis of types. "Whatever one might otherwise say about the energetic model, it at least has the function here of granting a discontinuity of pleasure, or of the *aisthēsis* in general, which would mean, by the same token, of the *aisthēsis* insofar as it is without any generality, or rather, insofar as it has only the dis-located generality, *partes extra partes*, not only *res extensa* in the Cartesian manner, but a general and generic being-outside-itself, a *zoned* being of the so-called 'sensuous' condition" ([Nancy 1996](), p. 16). Fourth, while *Listening* concentrates on *auditus*, this sense is treated by the physical qualities of timbre and effects of reverb straining toward a transient signification. Classical thought included timbre in arithmetical maps and was therefore deaf and mute, but for Nancy it is the sonorous matter that spreads and resounds in the reality of rhythm. Once sonorous beats become rhythm, that is, consistent cadence, we are in musical articulation ([Nancy 2007a](), p. 40).[15] Nonetheless, when Nancy folds matter in with undetermined physical arrangements, he propels "the remodeling of schemes of sonority (timbres, rhythms, notations) which itself is contemporaneous with the creation of a global sonorous space or scene... A musical-becoming of sensibility and a global-becoming of musicality have occurred, whose historiality remains to be thought about, all the more so since it is contemporaneous with an expansion of the image whose extent does not correspond to equivalent transformations in the perceptible realm" ([Nancy 2007a](), pp. 11–12). Fifth, deriving from the physical reverbs of erogenous zones possibly scheming representation, the audible is in debt to touch. "The qualitative indifference of the zones is exposed by touch: it is the endpoint of the process of stimulation (immediately after visual stimulation, Freud places that of touching, whereas he declared earlier that the former derives from the latter 'ultimately', as do not doubt all the others). Now, for all of tradition, touch, as we have already intimated, is nothing other than 'the sense of the body in its entirety', as Lucretius puts it. Touch is nothing other than the touch or stroke of sense altogether and of all the senses" ([Nancy 1996](), p. 17). Nancy's return to the body and the body's perpetual return to itself exposes the sonorous-haptic properties of the *corpse sonore*. Henceforth, we are in the realm of aesthetics where the primacy of touch is transimmanent; it plays in and in between the register of feeling and of thinking. However, Nancy's touch always touches at a distance. A proposition corresponding with the Aristotelean tradition. In *De Anima*, Aristotle gives primacy to the sense of touch as all other senses are understood as forms of tactility. Komel legibly explains Aristotle's touch as retaining "two different

registers: a metaphysical touch on the level of thinking, and a physical touch on the level of the body" (Komel 2016, pp. 116–17).[16]

**Tapping: Producing Rhythms.** When the sonorous strains toward sense on the premises of a univocal cause and the registers of understanding, we are in the realm of the tuned ear lending itself to the sound-image hanging between listening and the superiority of hearing of speculative music. *Listening* offers to disperse this musical *arche*, suggesting that if the materiality of the *corpse sonore* is no longer obliged to deduction, the connective tissue reverberating the conditional relations of its sensual zones lends to sonorous touches. I propose to practice this haptic proposition alongside artworks that make sonorous topologies by performing somatic projections. Although Nancy's musical references might seem uncanny in relation to his endeavour to free the sounding body from pregiven delineation, at least two of the musical references of his sonorous philosophy return to somatic sounds. In the first instance, the text pertains to Russian composer Igor Stravinsky and the second references Charles Rosen's book *The Frontiers of Meaning: Three Informal Lectures on Music.* For Rosen, music is distanced from signification since it acquires or assumes meaning (Rosen 1998, pp. 13, 126).[17] Musical meaning, or musical sense consolidates when aesthetic experience gradually reifies into tradition. Considered on its own premises and its particular relational context, that is, the relation of the singular plural, music may move toward signification; however, it can also move toward nonsense (Rosen 1998, pp. 11–12). Rosen thus asserts that "most attempts to attribute a specific meaning to a piece of music seem to be beside the point", since "the essential condition of music is its proximity to nonsense" (Rosen 1998, pp. 75, 125). The difference between understanding music logically or linguistically and feeling music aesthetically ties with Nancy's relational difference of hearing/listening as a manner of artistic creation. It also brings us back to the body, where Nancy finds Stravinsky's interest in somatic sounds:

> "When he was six years old, Stravinsky listened to a mute peasant who produced unusual sounds with his arms, which the future musician tried to reproduce: he was looking for a different voice, one more or less vocal than the one that comes from the mouth; another sound for another sense than the one that is spoken". (Nancy 2007a, p. 7)

Making somatic sounds by tapping, beating, bouncing, thumping, banging, and clapping, resonates soundscapes that are indeterminate and unpredictable. In what comes ahead, I elaborate on Nancy's sonorous-haptic proposition through listening and looking at Michael Snow's *Tap*. In 1969, Michal Snow experimented with dispersion and multiple durations by performing multiple rhythms, pulsating bodies. He performed tense intervals freed from a pregiven measured world to elaborate on the vibrating limits of dispersed somatic and technological pulsations. *Tap*, is an audiovisual composition composed of a tape player, a sound tape, a speaker, a wire, a framed text on paper and a black and white photograph. Each positioned in spatial detachment while echoing its others by way of sound, sight, and touch. Snow called this piece a dispersed composition. A heterogeneous opus that performs variable elements at once singular and relational. Their shared nature is contingent on our singular experience contingent on the plurality of relational sense; our visual, sonorous, haptic, conceptual, and perceptual reverberations. "I wanted to make a composition", wrote Snow,

> "which was dispersed, in which the elements would be come upon in different ways and which would consist of (1) a sound, (2) an image, (3) a text, (4) an object, (5) a line, which would be unified but the parts of which would be of interest in themselves if the connections between them were not seen (but better if seen). One of many considerations was that it be partly tactile, bodymade though using machines". (Snow [1969] 1994, p. 49)

*Tap* composes several rhythms. The work began with Snow tapping his fingers against a microphone. This is a first rhythm. He then moved the microphone over the tape recorder to produce another rhythm by feedback, after which he produced a third rhythm by looping

a selection of the recording. The selected drumming, as Snow called them, emanated from a dark brown loudspeaker, an *objet trouve*, "which spreads the 'created' element" (Snow [1969] 1994, p. 49). A fourth rhythm continued in the echo of a brown color rectangle framing a photograph resonating the speaker's color. The photo is a blowup featuring the recording reals as backdrop to the artist's fingers and mic. A fifth rhythm delineated the internal black echoes of the speaker's cable looping in the black and white reals and the text. In addition to these internal rhythms, it was significant to Snow to declare that the tape and photo were made in February 1969, while the text was typed in 14 March 1969. This announcement positions the work of articulating concepts and procedures within a spatio-temporal distance, while binding the activity of tapping to *écriture*. Hence, *Tap*'s *raison d'etre* was only delineated in post-reflection. The elements' tying inscriptions return to the body, since for Snow, "Typewriting is a very similar finger-tapping to the way the tape was made" (Snow [1969] 1994, p. 49). And this may be the sixth tempo forking out further rhythmical beats resonating internally and externally, amidst practices, technologies and senses.

*Tap*'s mosaic composite releases its organs to flat dispersion. Each organ in this composition is at once separate while connected to its others by sonorous, visual, and tangible echoes contingent on intervals that produce multiple rhythms. Intervals residing between the sensual senses amplified and altered by technologies, mic, recorder, speaker, cable, photograph, objects, and text, are released from their somatic source to produce other cadences. While Snow exhibits the different media of his tapping rhythms, he never exhibited the recording machine. It was an intentional decision that not only concealed the origin of the soundscapes but provided a motivating lack to propel dispersion and produce unexpected cadences that were contingent on relational intervals posed right at the limits, framing the limits, and connecting the work's internal organs. "'Rhythm' has its proper moment only in the gap of the beat that makes it into rhythm...", writes Nancy,

> "rhythm does not *appear*; it is the beat of appearing insofar as appearing consists simultaneously and indissociably in the movement of coming and going of forms or presence in general, and in the heterogeneity that spaces our sensitive or sensuous plurality. Moreover, this heterogeneity is itself as least double: it divides very distinct, incommunicable qualities (visual sonorous, etc.), *and* it shares out among these qualities other qualities (or the same ones), which one might name with 'metaphors'... but which are in the final analysis metaphors in the proper sense, effective transports or communication across the incommunicable itself, a general play of *mimesis* and of *methexis* mixed together across all the senses and all the arts". (Nancy 1996, p. 24)

Snow's *Tap*, his *corpse sonore*, performs the production of rhythms as a sonorous mosaic. The piece discards a fixed view or an a priori organizing principle in favor of performing a dispersed expanse which experiments with the artist's body, his fingers, producing sonorous topologies. The work does not only stretch our ears but also our minds. The sonorous nature of the piece steps out of its audible compartment to become an investigation of the visual and haptic properties of sonic materiality. The artist makes somatic sounds that expand and alter by using technological instruments to produce spatio-temporal dispersions. In doing so, *Tap* offers a sonic materiality, which, as Salomé Voegelin suggests, enables a "quasisonic consciousness contingent on the relational interdependence, the dynamic eventness of things, predicativeness, and duration" (Voegelin 2019, p. 560). Such sonorous sense is contingent on singular interruptions and relational effects as much as on rhythms occupying, appearing, dissolving, and becoming a heterogenous *corpse sonore*.

Fludd and Snow exhibit two ontological propositions: the first elaborates an exhibited origin from which stems a musical world order, the second performs somatic rhythms contingent on their multiple procedures to suggest multiple sonorous worlds. If Fludd's *Monochord* composes the world's form from a metaphysical first cause articulated as an effective interval, it hangs mid-air as a singular appearance of speculative music. Snow begins with a performance of his own body to effect others. However, his somatic rhythms

cannot be confined to materialism alone. For rhythm always necessitates a call for sense, a sense toward probable significations, always in the plural. Snow's sonorous event is hung mid-air; it does not regulate its performing organs, but allows to perform the heterogeneity of the senses and their amplified expressions. If we detach Fludd's *Monochord* from its truthful origin on the assumption of unprovable belief or the fear of ideological persuasion, if we overturn this musical diagram to begin our movement from the bottom up, from material soundings to conceptual reflection, that is, to making intangible sense, we are left with more than one sense, such that a singular manner of art no longer regulates being. If we take this proposition seriously, then we are no longer in the realm of art or music, but in the *corpse sonore* of the arts, of rhythms, of musics. *Listening* and *Tap* share the relational difference of the sonorous-haptic, expanding the senses while proposing to perform, experience, and conceptualize spaces and times in the plural [18]. "Therefore", writes Nancy, "the world is dis-located into plural worlds, or more precisely, into the irreducible plurality *of* the unity 'world': this is the a priori and the transcendental of art" (Nancy 1996, p. 18).

**Funding:** This research was supported by The Israel Science Foundation (grant No. 1730/18).

**Data Availability Statement:** The images published were sourced from the Wellcome Collection. Public Domain.

**Conflicts of Interest:** The author declares no conflict of interest.

## Notes

[1] "Here are the six categories of noises for the Futurist orchestra that we intend soon to realize mechanically: (1) roars, claps, noises of falling water, driving noises, bellows (2) whistles, snores, snorts (3) whispers, mutterings, rustlings, grumbles, grunts, gurgles (6) shrill sounds, cracks, buzzings, jingles, shuffles (5) percussive noises using metal, wood, skin, stone, baked earth etc. (6) animal and human voices: shouts, moans, screams, laughter, rattlings, sobs. . ." (Russolo 1986, p. 28).

[2] The concepts and practices of dynamism, simultaneity and interpenetration, expand beyond the scope of this essay. I note two Futurist sources: Boccioni et al. (1910), and Marinetti's writings on hapticity, "Tactilism: Toward the Discovery of New Senses", originally published in 1921, (Marinetti 2006, pp. 377–82).

[3] Nancy expands on Jacques Derrida's concept of "chantier" as the singular relational site of a song (chant) and a voice. In *Speech and Phenomena*, Derrida criticizes Hussel's use of the voice as bringing forth apparent transcendence. Instead, he offers auto-affection as the possibility of subjectivity contingent on the differential division of presence. Here, we may find the middle voice which precedes and sets up the opposition between passivity and activity. The middle voice that "speaks of an operation which is not an operation, which cannot be thought of either as a passion or an action of a subject upon an object, as starting from an agent or from a patient, or on the basis of, or in view of, any of these *terms*. But philosophy has perhaps commenced by distributing the middle voice, expressing a certain intransitiveness, into the active and the passive voice, and has itself been constituted in this repression". (Derrida 1973, p. 137). On Nancy's musical commentary see *Listening*, pp. 11–12.

[4] On Nancy's reverbs of Kant's aesthetic philosophy see *The Discourse of the Syncope: Logodaedalus* (Nancy 2008b). See also Louria Hayon (2022, pp. 174–208).

[5] Nancy imports Heidegger's double sense of *alētheia*, first, as an epistemological disclosure of truth (which is rejected), that is, unconcealment as a correspondence between an idea and a thing it represents. Second, *Alētheia* as ontological truth that designated disclosure itself. In "The Origin of the Work of Art" *alētheia* is the interplay between concealing and unconcealing. The latter rejects ontotheological projections and ontological calculations in favor of presence, that is, that the artwork is, and its unfathomable perceptual reserve, in "The Origin of the Work of Art" (Heidegger 1971, pp. 41, 66). See also, (Magnus 1970), on Nancy's critique of Heidegger's dualism see Ross (2007, pp. 134–63). On the relation of truth to Listening, see Janus (2011, p. 182).

[6] Snow's early work and Nancy's later writing concerning sonorous hapticity were practiced decades after the avant-garde's experimentation with newly organized sound. I am referring to Italian noises and Russian transrational practices which overturned the Platonic and Pythagorean approach to harmonious metaphysics morphed in music and existence. Russolo was not alone in his sonorous slapstick, avant-garde art from the early twentieth century includes many appearances clogging harmonious systems. These include experiments in *zaum* by Russian transrationalism and the new Futurist's typographies Parole in Libertà, Dada noises, Fluxus scores and more. To these I briefly note the shift propelling the empirical and unpredictable stance that appeared mid-20th century with John Cage's silence, chance operations, the prepared piano, and the acousmatic listening of *musique concrète* by Jérôme Peignot and Pierre Schaeffer's. See for example, Kahn (2001); and the recently published Weibel (2019). To note several works closer to Snow's *Tap* see for example, La Monte Young's Compositions (1960), his work with Marian

Zazeela *Dream* House (1969), Terry Riley's *In C* (1964), Nam June Paik's *Physical* Music (1964), Bruce Nauman's *Bouncing in a* Corner (1968–9), Alvin Lucier's *I am Sitting in a* Room (1969), and Hollis Frampton's film States (1969).

7    Brian Kane demonstrates how Nancy's use of the verb to listen plays between the French verbs 'entendre' and 'écouter.' Their difference is displayed by positing a comparison between Pierre Schaeffer and Nancy. In Schaeffer's phenomenological listening, entendre is related to subjective intentionality. Nancy tries to avoid this manner by giving primacy to écouter, which opens up the threshold between sense and signification to reveals the structure of resonance echoing the subject. "As the French makes explicit", Kane writes, "the struggle between sense and truth is a struggle between *écouter* and *entendre*. . . 'the subject is' 'a resonant subject' because both the object and the subject of listening, in his [Nancy] account, resonate. And they resonate because the object and subject of listening both share a similar 'form, structure or movement, that of *renvoi*. . . 'reference'. . . as both a sending-away (a dismissal) and a return. . . Both meaning and sound are comprised of a series of infinite referrals, a sending-away which returns, only to be sent away again, ever anew". In Kane (2012, pp. 442, 445).

8    Adrienne Janus elaborates on Nancy's anti-ocular turn in debt to Heidegger, Jacques Attali, Didier Anzieu and Peter Sloterdijk see (Janus 2011, pp. 182–91).

9    On Nancy's "musicking" see (Chapin and Clark 2013).

10   Janus asks why does Nancy fleetingly turns to noise and why does his "relative suppression of noise in his space of listening resemble a nineteenth-century concert hall? Why does he not make use of concepts associated with recent developments in music that would potentially be productive. . ." (Janus 2011, p. 200).

11   German musicologist Marius Schneider clearly defines the historical and anthropological symbolism of the speculative music portraying the cosmos as knowledge of harmonics: "The symbol is the ideological manifestation of the mythical rhythm of creation, and the degree of veracity attributed to the symbol is an expression of the respect which man is capable of according to this mythical rhythm". In Schneider (1957). See also, (Schneider 1959, pp. 39–62; Godwin 1989).

12   On speculation and creation as a reflection of the power of vision and intelligibility see (Rorty 1979, p. 13; Jay 1993, pp. 28–29).

13   Preceding Philippe-Rameau *Nouveau système de musique théorique* (1726), the fundamental bass is found in his 1722 *Treatise on Music*. See Christensen (2002, p. 54).

14   In Plato's *Timaeus*, Critias explains the forging of the world to Timaeus Hermocrates and Critias: "He began the division thus. First, he cut off one portion from the whole, next another, double of this. The third portion he made half as great again as the second, the thrice as great as the first, the fourth double of the second, the fifth three times the third, the sixth eight times, the seventh twenty-seven times the first. Then he proceeded to fill up the intervals of the double and the triple, still cutting off portions as before and inserting them in these intervals, so that in each interval there were two middle terms, the one exceeding and being exceeded by the same part of the extremes, the other exceeding and being exceeded by an equal number. These links gave rise to intervals of three to two, four to three, and nine to eight within the old intervals. So he filled up all the intervals of four to three with the interval of the nine to eight, leaving in each case a fraction such that the interval determined by it represented by the ratio 256 to 243. By this time, the blend from which he was cutting off these portions was last exhausted". See Plato (1929, pp. 35b–36b, 31–32). For commentaries on the harmonious structure of the world see: Macdonald (1937, pp. 59–116); Lippman (1964, pp. 1–44); McClain (1978, pp. 57–70); Pelosi (2010, pp. 68–89).

15   On Nancy's timbre see Janus, "Listening: Jean-Luc Nancy and the 'Anti-Ocular' Turn in Continental Philosophy and Critical Theory", pp. 191–92.

16   "That there is no sense in addition to the five—sight, hearing, smell, taste, touch—may be established by the following considerations:—We in fact have sensation of everything of which touch can provide us sensation (for all the qualities of the tangible *qua* tangible are perceived by us through touch); and absence of a sense necessarily involves absence of a sense-organ; and all objects that we perceive by immediate contact with them are perceptible by touch". *De Anima*, book 3, 425a15–20, *The Complete Works of Aristotle*, edited by Jonathan Barnes, vol. I (New Jersey: Princeton University Press, 1984), 675. On the separation of the Arts and their haptic contingency see Landes (2007, pp. 80–92). Finally and decisively, Jacques Derrida's account of the Aristotelean Nancy's and his haptic interruption, (Derrida 2005).

17   Interestingly Stravinsky's *Rite of* Spring (1913; the same year of Russolo's noise expositions) transformed classical rhythmic structure. The score experiments with tonality, metre, rhythm, stress, and dissonance. Interestingly, the piece was the third score composed for art critic and founder of the Ballets Russes Sergei Diaghilev. The first composition for *The Firebird* was premiered in 1910, the second was *Petrushka* in 1911. In 1915 Diaghilev convinced Stravinsky to visit Milan to hear Russolo's noise machines. "The evening in the salon of Marinetti—casa rossa, Corso Venezia, Milam—there was a meeting of the Futurist musicians, all of whom were present: Luigi Russolo, Balilla Pratella, Igor Stravinsky (who came especially from Lucerne), Prokofiev, Diaghilev (director of those Russian ballets that had become a choreographic epidemic), Massine (first ballerina), an exceptional Slavic pianist whose name construe who can, made up of difficult consonants, neither known nor written nor pronounced. . . the major attraction was Luigi Russolo with his twenty *Intonarumori*. Stravinsky wanted to have an exact idea of these bizarre new instruments, and, possibly insert two or three in the already diabolic scores of his ballets. . . A *crackler* crackled with a thousand sparkles like a fiery torrent. Stravinsky gushed emitting a syllable of crazy joy, leaped up from the couch of which he seemed a spring. Then a *rustler* rustled like petticoats of winter silk, like leaves in April, like sea rending summer. The frenzied composer

tried to find on the piano that prodigious onomatopoetic sound, in vein proved the semitones of his avid digits while the ballerina moved the legs of his craft". Payton (1976, pp. 28–29).

<sup>18</sup> "It stayed/Where I saw it/Then it moved a fraction/To the left and then twice that/Distance again further and fur-ther//It /Disappeared/Then just faintly/A corner of it just a fraction/Was visible if you peered/Very very closely/And just as/quietly/it was/ gone". Snow composed this poem in 1957. He was there to alter our perceptual manners, our cadence, our tangible relations to ourselves and to others. This piece, written from a distance of ocean and seas, is my adieu to a master artist who recently died.

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
