# Peer review of "Sonorous Touches: Listening to Jean-Luc Nancy’s Transimmanent Rhythms"

_arts, 2022_

Round 1

Reviewer 1 Report

- Check for spelling/editing errors throughout. I caught a few, e.g., Lns. 39, 68, 76, 274, and in FN 35 (*rite, *from).

- I recommend drawing some language from the paragraph at Ln. 157 for the abstract, to make your topic more readily understood and interesting for potential readers. 

Author Response

I wish to convey my gratitude to the reviewer for his/her efforts in reading the text and investing their valuable comments. Indeed, thank you for illuminating the requirement for language editing (which was completed by a language editor). Also, thank you for illuminating the requirement to elucidate the opening and end of the text, which I have revised.

For your convenience, pls see the highlighted revised sections.    

Again, many thanks for the reviewer's valuable comments. 

Reviewer 2 Report

This is an excellent article that competently addresses topics usually considered minor by current philosophical critics. I would only suggest that the author rework the conclusions, in the sense of trying to clearly present to the reader the implications of this essay. The feeling, perhaps even sought after by the author, is that the article ends ex abrupto, and I think that the article could be even more interesting if the conclusions were more definite and made explicit.

Author Response

I wish to convey my gratitude to the reviewer for his/her efforts in reading the text and investing their valuable comments. Indeed, thank you for illuminating that the text feels abruptly finished. Here, I have extended the conclusions, but must say that it was important for me to focus on the difference between Fludd's musical world order and Snow's dispersed sonorous worlds. The clarity of these two ontological propositions are important for me, while I am very cautious about imposing additional elaborations that might be risky to forceful. 

For your convenience, pls see the highlighted added sections.  

Again, many thanks for the reviewer's valuable comments. 
